# *SLC1A2* Gene Polymorphism Influences Methamphetamine-Induced Psychosis

**DOI:** 10.3390/jpm13020270

**Published:** 2023-01-31

**Authors:** Dayang Nooreffazleen Yahya, Rhanye Mac Guad, Yuan-Seng Wu, Siew Hua Gan, Subash C. B. Gopinath, Hasif Adli Zakariah, Rusdi Abdul Rashid, Maw Shin Sim

**Affiliations:** 1Department of Pharmaceutical Life Sciences, Faculty of Pharmacy, Universiti Malaya, Kuala Lumpur 50603, Malaysia; 2Department of Biomedical Science and Therapeutics, Faculty of Medicine and Health Science, Universiti Malaysia Sabah, Kota Kinabalu 88400, Malaysia; 3Department of Biological Sciences, School of Medical and Life Sciences, Sunway University, Subang Jaya 47500, Malaysia; 4Centre for Virus and Vaccine Research, School of Medical and Life Sciences, Sunway University, Subang Jaya 47500, Malaysia; 5School of Pharmacy, Monash University Malaysia, Bandar Sunway, Subang Jaya 47500, Malaysia; 6Faculty of Chemical Engineering and Technology, Universiti Malaysia Perlis, Arau 02600, Malaysia; 7Institute of Nano Electronic Engineering, Universiti Malaysia Perlis, Kangar 01000, Malaysia; 8Micro System Technology, Centre of Excellence (CoE), Universiti Malaysia Perlis (UniMAP), Pauh Campus, Arau 02600, Malaysia; 9Department of Psychological Medicine, Faculty of Medicine, Universiti Malaya, Kuala Lumpur 50603, Malaysia

**Keywords:** excitatory amino acid transporter, mania, methamphetamine dependence, psychosis, single nucleotide polymorphism

## Abstract

*SLC1A2* is a gene encoded for the excitatory amino acid transporter 2 which is responsible for glutamate reuptake from the synaptic cleft in the central nervous system. Recent studies have suggested that polymorphisms on glutamate transporters can affect drug dependence, leading to the development of neurological diseases and psychiatric disorders. Our study investigated the association of rs4755404 single nucleotide polymorphism (SNP) of the *SLC1A2* gene with methamphetamine (METH) dependence and METH-induced psychosis and mania in a Malaysian population. The rs4755404 gene polymorphism was genotyped in METH-dependent male subjects (*n* = 285) and male control subjects (*n* = 251). The subjects consisted of the four ethnic groups in Malaysia (Malay, Chinese, Kadazan-Dusun, and Bajau). Interestingly, there was a significant association between rs4755404 polymorphism and METH-induced psychosis in the pooled METH-dependent subjects in terms of genotype frequency (*p* = 0.041). However, there was no significant association between rs4755404 polymorphism and METH dependence. Also, the rs455404 polymorphism was not significantly associated with METH-induced mania for both genotype frequencies and allele frequencies in the METH-dependent subjects, regardless of stratification into the different ethnicities. Our study suggests that the *SLC1A2* rs4755404 gene polymorphism confers some susceptibility to METH-induced psychosis, especially for those who carry the GG homozygous genotype.

## 1. Introduction

Methamphetamine (METH) is an amphetamine derivative which is a potent and highly addictive stimulant that affects the central nervous system (CNS). The United Nations Office on Drugs and Crime (2019) [1] reported that METH is the second most abused drug in the world after cannabis, followed by opiates and cocaine. The occurrence is similarly seen in Malaysia. According to the statistics from the Malaysian National Anti-Drugs Agency (2019) [2], the number of METH users in Malaysia was approximately 6000 in 2014, and it increased by more than two-folds with ~16,000 cases reported in 2018. The increased number of METH users seen locally and globally is reportedly due to the expanding market of METH in the East, South–East Asia, and Oceania since 2010 (United Nations Office on Drugs and Crime, 2018) [3].

The *SLC1A2* (*EAAT2*) gene is located on chromosome 11p13 (National Centre for Biotechnology Information, 2020) [4]. It encodes a type of high-affinity glutamate transporter that is responsible for L-glutamate and D/L-aspartate uptake with cotransport of three sodium ions (Na+) and one proton (H+) and counter-transport of one potassium ion (K+) [5]. SLC1A2 is mainly expressed in astrocytes, neurons, and axonal terminals where it regulates glutamate concentrations in the CNS [6]. Therefore, this transporter is mechanistically important for regulating glutamate reuptake from the synaptic cleft in the CNS to maintain a low concentration of glutamate and prevent excitotoxicity [7] that can lead to neuron injury or death. SLC1A2 genes reuptake glutamate into neurons and catalyze them to form glutamine by glutaminase enzymes. Previous studies reported that an increase in glutamine was directly associated with psychotic symptoms and found that the astrocytic glutamate transporter is highly expressed in the prefrontal cortex of people with schizophrenia, which is relatively similar to drug-induced psychosis [8,9].

Many studies have reported that several classes of addictive drugs affect glutamate equilibrium and lead to the excessive release of glutamate that contributes to drug dependence [10,11]. The phenomenon is further confirmed when METH-dependent individuals show an altered level of glutamate metabolites in the brain that is associated with the dose and frequency of METH [12,13]. Furthermore, SLC1A2 gene variants are reported to be linked with alcohol dependence [14] which indicates their potential association with drug dependence.

To date, cocaine, amphetamine derivatives, opiates, and cannabinoids have been associated with glutamine neurotransmission [15]. However, the association between SLC1A2 gene polymorphism and drug dependence with drug abuse remains unknown. Recently, Veldic et al. (2019) discovered that the SLC1A2 gene is associated with a higher risk of pathological conditions related to changes in the extracellular glutamate levels [16]. Indeed, the SLC1A2 gene is considered a strong candidate for drug dependence and associated psychosis since its variants are hypothesized to be associated with schizophrenia, and schizophrenia’s symptoms, including paranoid hallucinations and delusions, resemble drug-induced psychosis [17,18].

Generally, it has been known that various populations differ in their genetic profiles and that there are variations in allele frequencies and genotypes among different ethnic groups. This phenomenon is attributed to the fact that genetic variability is affected in many aspects by peoples’ surrounding environments [19,20]. Therefore, our case-control study aims to establish the association of *SLC1A2* rs4755404 genetic polymorphism with METH-induced symptoms and dependence, which is a move forward regarding personalizing medicine in the Malaysian population.

## 2. Materials and Methods

### 2.1. Ethics Approval

The Medical Ethics Committee of Universiti Malaya Medical Centre (UMMC) has approved the study (MREC ID No.: 2017105-5641) which complies with the Declaration of Helsinki. All the drug-dependent subjects were allocated to drug rehabilitation centers for certain periods within which they were conscious during the interview session. Those who were not able to answer the questionnaire accurately were excluded. The subjects and their guardians received thorough explanations of the purpose of the study, and they could withdraw from study participation at any time.

### 2.2. Subject Recruitment and Sample Collection

Male subjects (*n* = 285) were recruited from a drug rehabilitation center that specializes in rehabilitating METH-dependent patients. The center is located in the Papar district, Sabah (East Malaysia). An internationally recognized questionnaire, the Mini-International Neuropsychiatric Interview (M.I.N.I.), was used for data collection. All subjects fulfilled the *Diagnostic and Statistical Manual of Mental Disorders, fifth edition (DSM-5*) criteria for amphetamine and METH dependence (American Psychiatric Association, 2013) [21]. Strict control was enforced to ensure the participants would not have any chance to be under METH influence or other drugs of abuse at the time of the study; thus, they were able to comprehend all the essential aspects of the study and its verbal questions. METH consumption was confirmed by a positive urine test during recruitment.

Qualified psychiatrists from UMMC (Associate Professor Dr. Rusdi Abdul Rashid and Professor Dr. Hatim Sulaiman) confirmed the presence of psychosis. Generally, the subjects were considered to have psychosis if they had delusions or presented with auditory, visual, or tactile hallucinations [22]. They were also considered to have mania if they fulfilled the DSM-V criteria for manic episodes based on a Mini-International Neuropsychiatric Interview [23]. Subjects with mixed or unclear ethnicity and those with significant medical illness or a history of psychiatric illness or those who were taking other drugs were excluded.

The controls were healthy volunteers recruited from two locations, including 1) the UMMC and 2) Luyang Health Clinic, Kota Kinabalu, Sabah. The controls were medically healthy, had no history of chronic and psychiatric disease, and did not fulfil the DSM-V criteria for amphetamine and METH dependence (American Psychiatric Association, 2013) [21]. The sample size was calculated by using the power for association with errors (PAWE), and 80% power was used for the study [24,25].

### 2.3. DNA Preparation and Analysis

In this study, all case subjects’ sampling was collected using a buccal swab, while most control subjects’ sampling was collected using blood samples (3 mL) in an ethylenediaminetetraacetic acid (EDTA) tube according to the standard protocol. DNA was extracted from buccal swab tissue and blood leukocytes using the QIAmp DNA Mini Kit (Qiagen, Hilden, Germany) and the QIAmp Blood Kit (Qiagen, Hilden, Germany), respectively. Genotyping of *SLC1A2* rs4755404 SNP was performed by using genotyping real-time polymerase chain reaction (qRT-PCR) with TaqMan^®^ SNP Genotyping Assay (Assay ID: C__27982507_10) and TaqMan^®^ RT Express Master Mix (Applied Biosystems, Foster City, CA, USA). Briefly, a final reaction (10 μL) was prepared which contained 1 μL of 10 ng/μL genomic DNA, 5 μL of 2X TaqMan™ Genotyping Master Mix, 0.5 μL of 20X Assay with VIC/FAM-labeled pre-designed probes, and 3.5 μL nuclease-free water. The thermal cycling conditions were as follows: 95 °C for 10 min, 40 cycles of 95° C for 15 sec, 60 °C for 1 min, followed by a final cycle of 60 °C for 1 min.

### 2.4. Statistical Analysis

Intergroup statistical analyses were performed using Chi-square and Fisher’s exact tests (with Bonferroni correction where appropriate). These analyses compared each ethnic group’s cases with ethnically matched healthy controls for the genotypic frequencies of rs4755404 C/C homozygous, C/G heterozygous, and G/G homozygous. Fisher’s exact tests were performed when the sample sizes were too small. In cases of multiple comparisons, the Bonferroni correction test was performed. The level of statistical significance was established as *p* < 0.05, and after Bonferroni correction for three subtests, the results were considered significant at *p* < 0.016.

## 3. Results

A total of 536 subjects participated in the study from multiethnic Malaysian populations, including the following: Malay = 185, Chinese = 98, Kadazan-Dusun = 144, and Bajau = 109 (Table 1). The study subjects were divided into two major groups—dependence vs. non-dependence groups. Further, the drug-use or dependent groups were stratified into psychosis vs. non-psychosis and manic vs. non-manic groups. The number of subjects should have been the same among psychosis and mania studies, but some of the psychotic subjects could not answer the questions accurately; therefore, they were excluded from further analysis to prevent inaccurate data, as suggested by professional psychiatrists. The mean age for the case subjects was 30.6 ± 8.0 years old, while the mean age for the control subjects was 31.8 ±10.3 years old.

The allelic and genotype frequencies for the polymorphism in the METH-dependent and control subjects are summarized in Table 2. The genotype distribution in both the METH-dependent subjects and the control subjects fulfilled the Hardy–Weinberg equilibrium. After stratifying into different ethnic groups, the genetic analysis of *SLC1A2* rs4755404 polymorphism did not have a significant association with METH dependence, as evident in both allele and genotype frequencies in all four ethnic groups investigated, which were Malay, Chinese, Kadazan-Dusun, and Bajau.

When genotype frequency was analyzed in the pooled METH-dependent subjects, it was found that it was highly associated with those who experienced psychosis compared with METH non-psychotic subjects. However, it was further discovered that such association observations were not dependent on ethnicity. No significant association was found for this polymorphism in both genotype and allele frequencies in all four ethnic groups (Table 3).

The results also showed that *SLC1A2* rs4755404 polymorphism was not markedly associated with METH-dependent subjects who had mania and those who did not have mania in all the ethnic groups after analyzing their genotype and allele frequencies. A comparison between the combination of different genotypes also showed no significant association between this polymorphism and METH-induced manic episodes (Table 4).

A significant difference was noted regarding the onset age of METH exposure among the four different ethnic groups (Table 5). Our finding showed that the Malaysian Chinese group has a later onset age than others at 30 ± 9.4 years old.

Additionally, we found a strong, significant association between the onset age of drug use and METH-induced mania, even after adjustment by the Bonferroni correction test (*p* = 0.002) (Table 6). However, no significant difference was found in the METH-induced psychosis group compared with the non-psychosis group (*p* = 0.810).

Furthermore, the effect of the *SLC1A2* rs4755404 polymorphism on the onset age of METH-induced symptoms and dependence was also analyzed (Table 7). There was no significant difference between the mean onset age and the genotypes in the METH-dependent groups. The mean was 24 ± 0.9 years for homozygous CC (*n* = 102), 25 ± 0.7 years for heterozygous CG (*n* = 130), and 24 ± 1.3 years for homozygous GG (*n* = 53). For METH-induced psychosis, our results showed that the means of onset age for homozygous GG (25 ± 10 years) and heterozygous CG (25 ± 8.0 years) were slightly higher compared with homozygous CC (23 ± 7.8 years), although no significant difference was found. Additionally, our results did not show any significant difference between the mean onset age for METH-induced mania and genotypes, which were approximately 20 years old.

## 4. Discussion

Studies have shown the involvement of glutamate transmission in drug dependence. However, only a few genetic polymorphisms of the *SLC1A2* gene have been successfully associated with or related to drug dependence. Presently, there are no studies that discuss *SLC1A2* rs4755404 genetic polymorphism and drug dependence. The present study investigated the genetic association between *SLC1A2* rs4755404 polymorphism and METH dependence and METH-induced symptoms, such as psychosis and mania.

This study assessed the association between *SLC1A2* rs4755404 polymorphism and METH dependence among the four ethnic groups in Malaysia, including Malay, Chinese, Kadazan-Dusun, and Bajau ethnic groups. Our findings showed that rs4755404 polymorphism did not significantly correlate with METH dependence in the pooled Malaysian subjects or after stratification into different ethnic groups. At present, there are no data available that report the association of this polymorphism with METH dependence. However, alcohol dependence was reported to be associated with this *SLC1A2* gene polymorphism in the German population [14]. Nevertheless, our results showed that the Malay group carried ‘C’ allele and had a 1.533 times higher risk of developing METH dependence compared with those with ‘G’ allele, although the *p*-value was not significantly different. Hence, a further study should be conducted with a larger sample size to confirm the possible risk of developing METH dependence for case subjects carrying ‘C’ allele.

A significant association was found between rs4755404 polymorphism and METH-induced psychosis, which was evident in the genotype frequencies in the pooled subjects. Interestingly, this result is consistent with the study by [18] which reported that *SLC1A2* gene variants were associated with psychotic symptoms, such as delusions and hallucinations, which are similar to METH-induced psychosis.

However, after stratification into different ethnic groups, no significant association was observed for this polymorphism in terms of genotype and allele frequencies in the METH-dependent subjects who had psychosis compared with those who did not. Interestingly, we further found that the homozygous C/C genotype occurred more often than the heterozygous C/G and homozygous G/G genotypes in the Bajau group, which was not observed in other ethnic groups. This study also demonstrated that the Bajau group carrying ‘C’ allele had a 2.161 times higher risk of developing METH-induced psychosis compared with ‘G’ allele carriers, although the *p*-value was not significantly different. These results demonstrated that the risk of developing METH-induced psychosis in the pooled METH subjects carrying the homozygous C/C genotype was significantly higher (2.199 times the odds) than those carrying the homozygous G/G genotype.

As mentioned, genetic variants of the EAAT2 gene (SLC1A2) have been suggested to be related to schizophrenia as some psychotic symptoms, such as paranoid hallucinations and delusions, are similar to those of drug dependence, which suggests that the EAAT2 gene is a strong candidate for drug dependence and its related psychosis. Moreover, a significant association between an SLC1A2 SNP, the synonymous G603 A (rs752949) in exon 5, has also suggested the susceptibility of this gene to risk-taking behaviour in alcohol dependence. These findings provide evidence to support the involvement of the EAAT2 gene in aspects of drug dependence [17]. Additionally, SLC1A2 polymorphism was shown to significantly influence other mental health disorders related to the total episode recurrence rate in bipolar disorders, with G homozygotes reporting the highest number (*p* = 0.004) [26]. Therefore, we hypothesized that this genotype’s effects could be linked to a possible interaction between METH and glutamatergic systems that causes alteration in the synaptic expression of glutamate receptor 2 related to the findings in the current study.

It was also found that the risk of developing METH-induced psychosis in those who carry homozygous C/C plus heterozygous C/G was 2.303 times higher than those who carry homozygous G/G in the pooled METH subjects. After stratification into different ethnic groups, those with homozygous C/C plus heterozygous C/G in the Bajau group had a significantly higher risk (8.919 times the odds) to develop METH-induced psychosis compared with those with homozygous G/G. Therefore, this may suggest that the ‘C’ allele carrier is vulnerable to METH-induced psychosis, and ‘G’ allele may have a protective role against METH-induced psychosis in the METH-dependent subjects.

METH use has been highly linked with manic episodes [27]. However, our study found no significant association of the *SLC1A2* rs4755404 gene polymorphism in both genotype and allele frequencies in METH-dependent subjects with or without mania, either in the pooled subjects or the ethnic groups studied. Thus, the *SLC1A2* rs4755404 gene polymorphism may not have any association with METH-induced mania.

In this study, we found a strong significant difference in the onset age of METH abuse and dependence among the four ethnic groups in Malaysia. This significant difference may be due to the late-onset age of METH abuse and dependence in the Chinese population (30 ± 9 years) compared with other ethnic groups that had earlier onset ages of approximately 22–23 ± 8 years. Furthermore, we also analyzed ages with the onset of METH-induced symptoms, such as psychosis and mania, among the METH users compared with those without symptoms. Interestingly, a significantly higher risk of developing METH-induced mania was found when the subjects were exposed to and became dependent on METH at a younger age. Moreover, the *p*-value remained significant following the Bonferroni correction. The results indicated that early-onset METH users (those who began using at age twenty or younger) were more likely to develop manic episodes after METH was used than late-onset users of METH. A similar trend was observed in METH-induced psychosis subjects, although the *p*-value was not significantly different.

Besides that, the effect of the *SLC1A2* rs4755404 genetic polymorphism on the onset age of METH abuse was also analyzed in the METH-dependent subjects. Unfortunately, our findings did not show any significant association between genotypes and the onset age of METH dependence, psychosis, and mania. This phenomenon may be compatible with a previous study that did not indicate an association of *SLC1A2* polymorphisms with the onset of schizophrenia, which is similar to METH-induced psychosis [28].

Although this study found some significant associations, it still has some limitations. One of the limitations is that the other genetic polymorphisms of the *SLC1A2* gene were not investigated. This study only focused on rs4755404 polymorphism. Other *SLC1A2* polymorphisms should be investigated as they may contribute to the risk of developing METH dependence and METH-induced psychosis and manic episodes. Another limitation is that the subjects were all males. However, METH use disorder is more prevalent among males than females, which provides greater homogeneity in the study population. Lastly, the sample size in our study was small and included less evidence to confirm our finding that METH-induced symptoms and dependence were influenced by *SLC1A2* rs4755404 polymorphism. Intermarriages between ethnicities are also common in Sabah; therefore, future studies should thoroughly investigate family backgrounds to exclude mixed populations from the study as the SNPs in this study were significantly influenced by the mixed population.

## Figures and Tables

**Table 1 jpm-13-00270-t001:** Demographic data for METH dependent and control subjects.

Characteristics	Malay (*n* = 185)		Chinese (*n* = 98)		Kadazan-Dusun (*n* = 144)		Bajau (*n* = 109)		Total (*n* = 536)	
	Case (*n* = 95)	Control (*n* = 90)	Case (*n* = 36)	Control (*n* = 62)	Case (*n* = 79)	Control (*n* = 65)	Case (*n* = 75)	Control (*n* = 34)	Case (*n* = 285)	Control (*n* = 251)
Age, mean (SD), year	30.94 (7.4)	32.0 (9.8)	38.3 (9.2)	31.2 (9.2)	29.3 (6.7)	31.9 (11.2)	27.9 (6.9)	33.7 (12.1)	30.6 (8.0)	31.8 (10.3)
*METH-dependent subjects:*										
With psychosis, *n* (%)	55 (0.579)		15 (0.416)		26 (0.329)		23 (0.307)		119 (0.418)	
Without psychosis, *n* (%)	40 (0.421)		21 (0.584)		53 (0.671)		52 (0.693)		166 (0.582)	
	Case (*n* = 70)		Case (*n* = 34)		Case (*n* = 58)		Case (*n* = 65)		Case (*n* = 227)	
Age, mean (SD), year	31.3 (8.2)		38.6 (9.4)		28.2 (6.6)		27.8 (7.0)		30.6 (8.0)	
*METH-dependent subjects:*										
With mania, *n* (%)	12 (0.171)		6 (0.176)		12 (0.207)		8 (0.123)		38 (0.167)	
Without mania, *n* (%)	58 (0.829)		28 (0.824)		46 (0.793)		57 (0.877)		189 (0.833)	

Abbreviations: SD, standard deviation.

**Table 2 jpm-13-00270-t002:** Genotype and allele frequencies of the *SLC1A2* rs4755404 polymorphism in male controls and male METH-dependent subjects.

Ethnicity	Subject	Genotype, *n* (Frequency)			*p*-Value	Allele, *n* (Frequency)		*p*-Value	OR (95% CI)
		C/C	C/G	G/G		C	G		
Malay	Cases	35 (0.368)	47 (0.495)	13 (0.137)	0.115	117 (0.616)	73 (0.384)	0.054	1.533 (1.014–2.318)
	Controls	24 (0.267)	44 (0.489)	22 (0.244)		92 (0.511)	88 (0.489)		
Chinese	Cases	11 (0.305)	20 (0.556)	5 (0.139)	0.606	42 (0.583)	30 (0.417)	0.798	0.884 (0.489–1.598)
	Controls	24 (0.387)	28 (0.452)	10 (0.161)		76 (0.613)	48 (0.397)		
Kadazan-Dusun	Cases	22 (0.278)	38 (0.481)	19 (0.241)	0.929	82 (0.519)	76 (0.481)	0.833	0.925 (0.581–1.472)
	Controls	20 (0.308)	30 (0.461)	15 (0.231)		70 (0.538)	60 (0.462)		
Bajau	Cases	34 (0.453)	25 (0.333)	16 (0.213)	0.505	93 (0.620)	57 (0.380)	0.981	0.949 (0.524–1.716)
	Controls	14 (0.412)	15 (0.441)	5 (0.147)		43 (0.632)	25 (0.368)		
Total	Cases	102 (0.358)	130 (0.456)	53 (0.186)	0.700	334 (0.586)	236 (0.414)	0.422	1.113 (0.873–1.419)
	Controls	82 (0.327)	117 (0.466)	52 (0.207)		281 (0.560)	221 (0.440)		

Abbreviations: CI, confidence interval and OR, odds ratio. Note: the C allele is assumed as the exposed group.

**Table 3 jpm-13-00270-t003:** Genotype and allele frequencies of the *SLC1A2* rs4755404 polymorphism in male METH-dependent subjects with and without psychosis.

Ethnicity	Subject	Genotype, *n* (Frequency)			*p*-Value	Allele, *n* (Frequency)		*p*-Value	OR (95% CI)
		C/C	C/G	G/G		C	G		
Malay	Psychosis	21 (0.382)	28 (0.509)	6 (0.109)	0.653	70 (0.636)	40 (0.364)	0.594	1.229 (0.681–2.218)
	No psychosis	14 (0.350)	19 (0.475)	7 (0.175)		47 (0.588)	33 (0.412)		
Chinese	Psychosis	4 (0.267)	9 (0.600)	2 (0.133)	0.894	17 (0.567)	13 (0.433)	1.000	0.889 (0.344–2.298)
	No psychosis	7 (0.333)	11 (0.524)	3 (0.143)		25 (0.595)	17 (0.405)		
Kadazan-Dusun	Psychosis	8 (0.308)	13 (0.500)	5 (0.192)	0.770	29 (0.558)	23 (0.442)	0.608	1.261 (0.647–2.456)
	No psychosis	14 (0.264)	25 (0.472)	14 (0.264)		53 (0.500)	53 (0.500)		
Bajau	Psychosis	12 (0.522)	10 (0.435)	1 (0.043)	0.054	34 (0.739)	12 (0.261)	0.069	2.161 (1.007–4.639)
	No psychosis	22 (0.424)	15 (0.288)	15 (0.288)		59 (0.567)	45 (0.433)		
Total	Psychosis	45 (0.378)	60 (0.504)	14 (0.118)	**0.041**	150 (0.630)	88 (0.370)	0.083	1.371 (0.975–1.928)
	No psychosis	57 (0.343)	70 (0.422)	39 (0.235)		184 (0.554)	148 (0.446)		

Abbreviations: CI, confidence interval and OR, odds ratio. Note: boldface shows a significant difference with a *p*-value less than 0.05. The C allele is assumed as the risk exposed group.

**Table 4 jpm-13-00270-t004:** Odds ratio in various genetic models for the different ethnic groups in METH dependence, METH-induced psychosis, and mania.

Symptom	Genotype	Malay		Chinese		Kadazan-Dusun		Bajau		Total	
		Odds Ratio (95% CI)	*p*-Value	Odds Ratio (95% CI)	*p*-Value	Odds Ratio (95% CI)	*p*-Value	Odds Ratio (95% CI)	*p*-Value	Odds Ratio (95% CI)	*p*-Value
METH dependence	CC vs. GG	2.468 (1.044–5.833)	0.062	0.917 (0.253–3.327)	0.843	0.868 (0.350–2.154)	0.942	0.759 (0.233–2.473)	0.869	1.220 (0.755–1.973)	0.490
	CC vs. (CG + GG)	1.604 (0.868–3.000)	0.185	0.697 (0.291–1.670)	0.553	0.868 (0.422–1.785)	0.842	1.185 (0.522–2.691)	0.844	1.149 (0.803–1.643)	0.504
	(CC + CG) vs. GG	2.041 (0.957–4.352)	0.093	1.192 (0.373–3.811)	0.995	0.947 (0.437–2.054)	0.952	0.636 (0.212–1.907)	0.582	1.144 (0.746–1.753)	0.611
METH- induced psychosis	CC vs. GG	1.750 (0.485–6.314)	0.595	0.857 (0.098–7.510)	0.676	1.600 (0.419–6.114)	0.724	8.182 (0.960–69.750)	0.066	**2.199** **(1.065–4.541)**	**0.048**
	CC vs. (CG + GG)	1.147 (0.492–2.677)	0.919	0.727 (0.169–3.133)	0.951	1.238 (0.441–3.477)	0.890	1.488 (0.555–3.987)	0.589	1.163 (0.713–1.898)	0.632
	(CC + CG) vs. GG	1.732 (0.534–5.617)	0.535	1.083 (0.158–7.435)	0.684	1.508 (0.477–4.765)	0.673	**8.919** **(1.101–72.248)**	**0.037**	**2.303** **(1.187–4.470)**	**0.019**
METH- induced mania	CC vs. GG	0.947 (0.150–5.994)	0.677	0.444 (0.022–9.032)	0.788	0.619 (0.089–4.316)	1.000	2.308 (0.236–22.598)	0.812	1.038 (0.381–2.824)	0.856
	CC vs. (CG + GG)	2.053 (0.584–7.218)	0.422	0.422 (0.043–4.165)	0.794	0.457 (0.088–2.364)	0.557	1.192 (0.271–5.241)	0.884	0.925 (0.445–1.926)	0.982
	(CC + CG) vs. GG	0.577 (0.102–3.279)	0.898	0.833 (0.076–9.129)	0.627	1.182 (0.276–5.067)	0.891	2.500 (0.284–22.043)	0.681	1.114 (0.456–2.725)	0.988

Abbreviation: CI, confidence interval. Note: boldface shows a significant difference with *p*-values less than 0.05. The C allele is assumed as the exposed group.

**Table 5 jpm-13-00270-t005:** The age of onset for METH exposure and dependence.

Ethnicity	*n*	Mean Onset Age (Year)	Standard Deviation	*p*-Value
Malay	95	23.9	8.8	**<0.001** ^†^
Chinese	36	30.2	9.4	
Kadazan-Dusun	79	23.5	7.9	
Bajau	75	22.0	8.1	

Note: boldface shows a significant difference with *p*-values less than 0.05. ^†^ shows a significant difference after adjustment by the Bonferroni correction test.

**Table 6 jpm-13-00270-t006:** The age METH users developed METH-induced psychosis and mania.

Symptoms	Mean Age (Year)	Standard Deviation	Standard Error	*p*-Value
Dependence without psychosis (*n* = 166)	24.2	9.1	0.7	0.810
Dependence with psychosis (*n* = 119)	23.9	8.2	0.8	
Dependence without mania (*n* = 189)	24.7	9.2	1.0	**0.002** ^†^
Dependence with mania (*n* = 38)	19.9	6.2	0.7	

Note: boldface shows a significant difference with *p*-values less than 0.05. ^†^ shows a significant difference after adjustment by the Bonferroni correction test.

**Table 7 jpm-13-00270-t007:** The age of onset for METH dependence, METH-induced psychosis, and mania, according to genotypes.

Symptoms	Genotype	*n*	Mean Onset Age (Year)	Standard Deviation	*p*-Value
METH dependence	Homozygous C/C	102	23.7	0.9	0.793
	Heterozygous C/G	130	24.5	0.7	
	Homozygous G/G	53	23.9	1.3	
METH-induced psychosis	Homozygous C/C	45	22.6	7.8	0.376
	Heterozygous C/G	60	24.7	8.0	
	Homozygous G/G	14	25.0	10.3	
METH-induced mania	Homozygous C/C	13	20.3	6.6	0.813
	Heterozygous C/G	18	19.2	5.7	
	Homozygous G/G	7	20.9	7.5	

## Data Availability

Not applicable.

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
