# Peer review of "SLC1A2 Gene Polymorphism Influences Methamphetamine-Induced Psychosis"

_jpm, 2023, doi:10.3390/jpm13020270_

Round 1

Reviewer 1 Report

The review of the manuscript entitled: “SLC1A2 gene polymorphism influences methamphetamine-induced psychosis

Comments for Authors:

Thank you for the valuable research you have done. The study concerns favorable topic and acceptable writing. However, there are some issues:

1) ‘Materials And Methods’ section, ‘Ethics approval’ sub-section, the authors mentioned: “A written informed consent was obtained from each subject prior to the study enrollment”. However, manic or psychotic patients may not have enough capacity for decision making.

2) ‘Materials And Methods’ section, ‘DNA Preparation and Analysis’ sub-section, the authors said: “Should the subjects refuse to provide blood samples, buccal swab tissues were collected for DNA extraction using a QIAmp DNA Mini Kit (Qiagen, Germany) which gives similar yield”. However, the authors did not mentioned, the exact number of cases who refused blood sampling. Furthermore, if buccal swab gives similar yield; why the authors used more aggressive way of blood sampling for the other participants?

3) In ‘Table 1’, the total number of cases for psychosis or mania is confusing for readers. For example, in the last column, the total number of cases with and without mania is 227 which is less than all psychotic and non-psychotic cases (285) and more than non-psychotic cases (166). Is this because of excluding manic patients who showed psychotic features simultaneously?

4) It is recommended for authors to mention about non-equal number of participants from each ethnicity in limitations of the study. The effect would be more prominent in results for age onset of Chinese (Table 5).

Good luck

Author Response

We thank the reviewer for the valuable comments and suggestions. We appreciate it and have now responded to it as per attachment 

Reviewer 2 Report

excellent paper & contribution to science 

please mention the implication of these genes 

Author Response

We thank the reviewer for the valuable comment. We have now incorporated it to our manuscript
